# The changing face of desktop video game monetisation: An exploration of exposure to loot boxes, pay to win, and cosmetic microtransactions in the most-played *Steam* games of 2010-2019

David Zendle[1]*, Rachel Meyer[1], Nick Ballou[2]

**1** University of York, York, United Kingdom, **2** Queen Mary University of London, London, United Kingdom

* david.zendle@york.ac.uk

**Data Availability Statement:** All relevant data are held openly on the OSF repository located at https://osf.io/wpqx7/.

## Abstract

It is now common practice for video game companies to not just sell copies of games themselves, but to also sell in-game bonuses or items for a small real-world fee. These purchases may be purely aesthetic (cosmetic microtransactions) or confer in-game advantages (pay to win microtransactions), and may also contain these items as randomised contents of uncertain value (loot boxes). The growth of microtransactions has attracted substantial interest from both gamers, academics, and policymakers. However, it is not clear either how frequently exposed players are to these features in desktop games, or when any growth in exposure occurred. In order to address this, we analysed the play history of the 463 most-played *Steam* desktop games from 2010 to 2019. Results of exploratory joinpoint analyses suggested that cosmetic microtransactions and loot boxes experienced rapid growth during 2012–2014, leading to high levels of exposure by April 2019: 71.2% of the sample played games with loot boxes at this point, and 85.89% played games with cosmetic microtransactions. By contrast, pay to win microtransactions did not appear to experience similar growth in desktop games during the period, rising gradually to an exposure rate of 17.3% by November 2015, at which point growth decelerated significantly (p<0.001) to the point where it was not significantly different from zero (p = 0.32).

## Introduction

The way that the video game industry makes money has undergone important changes in recent decades. In the 1990s and early 2000s, industry profits were largely based around the sale of copies of games [1]. These copies might take the form of cartridges, discs, or even digital downloads. Under this model individuals were handing over money in return for the either the ownership of a complete product, or the license to play that product for a potentially unlimited period of time [2]. Similarly, ownership of a product might occur via a subscription-

**Funding:** Nick Ballou is a PhD student whose doctoral training is funded by the EPSRC Centre for Doctoral Training in Intelligent Games & Game Intelligence (IGGI).

**Competing interests:** The authors have declared that no competing interests exist.

based model: It has been common for decades for players of online role-playing games to pay a flat monthly charge for access to a game.

However, at some point in the early 2000s, monetisation in video games underwent a significant shift. As well as selling games as complete products, publishers also began offering gamers the ability to purchase additional items, bonuses or services within the game itself for a real-money fee, known as a 'microtransaction' [3].

## Cosmetic microtransactions

As noted in [4], many microtransactions allow players to purchase decorations and alternative costumes that "offer no in-game advantage and are purely aesthetic". In the context of this paper, we refer to any situation in which spending additional money leads to an aesthetic change within a game but no in-game advantage as a 'cosmetic microtransactions'.

The cosmetic microtransactions that may be made in video games are varied. For example, in the multiplayer battle royale game *Fortnite*, players can spend real-world money to buy in-game 'emotes' that allow them to express ideas and feelings via the movements of their in-game avatar. In the vehicular soccer game *Rocket League*, players can pay to purchase new 'goal explosions' that allow them to celebrate in-game victory with unique visual effects. And in the third-person shooting game *Anthem*, players can buy new armour pieces for their in-game mechanical suits of armour. These pieces do not confer any in-game boosts or advantages in terms of fighting: They simply look different.

## Pay to win microtransactions

However, as noted in [5], not all microtransactions in games are purely cosmetic in nature. Players of many modern video games are also given the option to purchase virtual items and bonuses that increase their chances of in-game success. In this paper, we define any situation in which players are able to exchange real-world money for something that increases their chance of in-game success as a 'pay to win' microtransaction.

Some 'pay to win' microtransactions do not have any effect on the aesthetic of a game. For example, players of the multiplayer mode in *The Last of Us* can pay real-world money for advantages such as the ability to sneak up on other players silently via an 'Agility perk'. This in-game advantage does not change how the game itself looks: It merely alters how the game is played.

However, other "pay to win" microtransactions also change how a game looks. For example, players of the game *Awesomenauts* can spend real-world money to purchase additional in-game characters. These new characters can convey an in-game advantage. However, they also have unique and special looks, and therefore have cosmetic value as well. Within the context of this paper, if a microtransaction changes both how a game looks and confers an in-game advantage, we would categorise that microtransaction as 'pay to win'.

It is import to note that some games separately offer both cosmetic microtransactions and pay to win microtransactions. An example of this is *Assassin's Creed*: *Odyssey*. In this game, players may pay real-world money to purchase a boost that enables them to level up more quickly, but does not change how the game itself looks. This is a pay to win microtransaction. Conversely, they may spend real-world money to purchase a 'skin' for their in-game mount that changes how it looks, but does not affect gameplay–this is an entirely cosmetic microtransaction.

Pay to win microtransactions are thought to have originated with online multiplayer games such as *MapleStory* in the early 2000s [6], and have garnered controversy amongst both gamers and academics alike. Criticisms of pay to win microtransactions are wide-ranging. Some

academics provide ethical critiques of how they may change "the game from a competition where the best player wins to . . . who wants to and can pay the most" [7]; others posit a belief that this model makes games unfair for less affluent players [8]. In [9], researchers suggest that they may encourage the entrapment of players. They posit that games such as *Candy Crush* may set up situations in which in-game goals are almost attained ('near misses') in order to encourage pay to win purchasing; and that this strategy may lead to continued play and spending "to the similar extent of wins as demonstrated in the gambling literature".

Controversies over pay to win have led some game developers to explicitly reject these microtransactions as an element of their design philosophies [10]. Furthermore, despite the popularity of games with pay to win elements, many individuals have publicly voiced their displeasure with their incorporation in the games that they play [11].

## Loot boxes

As described above, microtransactions can lead to both aesthetic differences, and gameplay advantages. However, when making a purchase, players are not always aware what advantage or difference they are buying due to a monetisation strategy known as loot boxes. A definition of loot boxes is given in [12] as follows:

1. Loot boxes are items in video games that may be bought for real-world money, but which provide players with a randomized reward of uncertain value

This may be considered a restrictive definition of loot boxes: For example, a sceptic might suggest that in-game items rewarded purely through gameplay be considered loot boxes. We would defend the above definition on the basis that it explicitly invokes the potential for monetisation, which sits at the heart of many issues regarding loot boxes. Furthermore, it was used in oral testimony to a recent UK Parliamentary Select Committee, and was subsequently used by this legislative subcommittee in their official report regarding the potential for harm present in loot boxes [13, 14]. It therefore provides a widely-used and useful definition of loot boxes. This definition is used throughout this paper.

Loot boxes take diverse forms. Some may be considered pay to win: For example, players of the fighting game *Marvel*: *Contest of Champions* may pay real-world money to open sealed in-game crystals that contain characters from *Marvel* franchises. Owning powerful and rare characters can help the player win in-game fights. However, when a player hands over their money to open a crystal, they have no way of knowing whether the character that crystal contains is a rare and powerful one, or a weak and common one.

Others loot boxes may be considered purely cosmetic microtransactions. For example, players of *Counter-Strike*: *Global Offensive* may spend real world money to unlock sealed 'weapon cases'. Each case contains a novel aesthetic for an in-game gun or knife. However, when paying to open a weapon case, players do not know which cosmetic upgrade they are paying for.

Loot boxes are thought to be extraordinarily lucrative for the video games industry, with one source estimating that they may have generated as much as $30 billion in revenue in 2018 alone [15]. However, there are distinct concerns about this monetisation strategy. As noted in [16], loot boxes share distinct similarities with gambling. This has led to concerns that engaging with loot boxes may lead to increases in gambling amongst gamers [17]. Evidence for this causal mechanism is unclear. Spending on loot boxes has been repeatedly linked to problem gambling. However, it is uncertain whether this is because loot boxes cause problem gambling, or whether it is because individuals with pre-existing gambling problems spend more money on loot boxes [18–20].

### The present research

It is widely acknowledged that both pay to win microtransactions, cosmetic microtransactions, and loot boxes have become more common in recent years. This has been accompanied by substantial interest.

However, how these features are changing over time is unclear. For example, some news reports have recently suggested that loot boxes are currently becoming more widespread [21], whilst others report that loot boxes are currently in decline [22]. Still more imply that the prevalence of specific in-game features may render them relatively unimportant: A recent statement from one industry representative characterises loot boxes as "a particular form of randomised in-game purchase which feature[s] in a minority of games" [23]. However, to the best of our knowledge no piece of academic research has investigated changes in exposure to either loot boxes, pay to win microtransactions, or cosmetic microtransactions.

This piece of research therefore sets out to explore the changing rate of exposure to loot boxes, pay to win microtransactions, and cosmetic microtransactions by analysing historical data on how many individuals play games with these features each day.

The *Steam* platform is often considered to be the dominant way for desktop video games to be both sold and delivered [24, 25]. In this piece of research, we create a dataset of the number of players of each of the most-played *Steam* games. This dataset records the peak number of simultaneous players for each game on each day from the 22$^{nd}$ March 2010 to the 22$^{nd}$ April 2019. We then code each of these games for the presence of loot boxes, pay to win microtransactions, and pay to win microtransactions. We then explore how these features change within the sample over time via a joinpoint analysis.

## Method

### Ethics

This research consisted of an analysis of SteamDB data. SteamDB is a publicly available database that lists the daily number of players of a variety of games. For example, the daily number of players of Counter-Strike can be viewed by browsing to https://steamdb.info/app/730/.

This study solely makes use of SteamDB data. Due to the publicly available and naturally anonymous nature of the aggregate data used in this study, ethical approval was not applied for when conducting this study. Upon submission of this manuscript, a formal waiver from the lead author's host institution was requested by journal staff. Said waiver was applied for and granted by the ethics officer for the lead author's department on the basis that this project uses a publicly available database of information that is not personally identifiable.

SteamDB is a browsable database of aggregate play data from a variety of desktop games. The design and conduct of this research did not violate the terms and conditions of SteamDB.

### Design

A list was made of the all-time most-played desktop games on the *Steam* platform. This was operationalised as any game that had achieved over 10,000 simultaneous players. This led to the creation of a list of 474 games that fit this criterion on the 22$^{nd}$ April 2019 via reference to the *SteamDB* website [26], which keeps a record of the peak number of simultaneous players for each game on the *Steam* platform.

The complete play history of each of these games was then extracted in turn from *SteamDB*. Inspection of these records revealed that a daily log of peak simultaneous players was kept for each game by the *Steam* platform from 22$^{nd}$ March 2010.

This process was achieved by first navigating a browser to the *SteamDB* website, where a list of games ordered by all-time peak simultaneous players is displayed. All entries with 10,000 or more simultaneous players were noted down by a researcher. The researcher then used a browser to navigate to the SteamDB page for each of the 474 games. A link on each of these pages allowed the direct download of a.csv file containing the complete history of the number of players of that game.

## Measures

The following three variables were then measured for each of these games:

1. **The presence of loot boxes,**

2. **The presence of pay to win microtransactions**

3. **The presence of cosmetic-only microtransactions.**

**The presence of loot boxes.**    Using the definition of loot boxes given earlier, coders were instructed to record that a game tested positive for the presence of loot boxes if it contained in-game items that could be bought for real-world money but which contained randomized rewards. An example of a game that would test positive for loot boxes is *NBA2K18*, a basketball game in which gamers can pay real-world money to purchase 'player packs' that contain a randomised assortment of new basketballers for their team. An example of a game that would test negative for loot boxes is *The Elder Scrolls*: *Oblivion*. Players of this game could pay real-world money to purchase new in-game content (for example, armour for their horses). However, when handing over their money they always knew what they would get in return.

**The presence of pay to win microtransactions.**    Using the definition of pay to win microtransactions given earlier, coders were instructed to record that a game tested positive for the presence of pay to win microtransactions if players could pay real-world money to in any way increase their chances of in-game success. An example of a game that would test positive for pay to win microtransactions is *Grand Theft Auto V*, in which players may pay real-world money for in-game currency, that can be used to purchase powerful new weapons. A game that would test negative for pay to win microtransactions is the team-based strategy game *DOTA 2*. In this game, players can pay real-world money to unlock new aesthetics for their in-game characters: However, spending money can never confer an in-game advantage.

**The presence of cosmetic microtransactions.**    Using the definition of cosmetic microtransactions 2given earlier, coders were instructed to record that a game tested positive for the presence of cosmetic microtransactions if players could pay real-world money for things that offered no in-game advantage and purely led to an aesthetic change. An example of a game that would screen positive for cosmetic microtransactions is *Rocket League*, in which a variety of decals, goal explosions, and other aesthetic effects may be bought for real-world money. However, none of these purchases are able to change how the game is played. A game that would test negative for cosmetic microtransactions is the digital collectible card game *Artifact*, in which players can pay real-world money to purchase new cards, all of which have some theoretical gameplay value.

It should be noted that some games could be coded as containing both cosmetic and pay to win microtransactions: A good example of this is *Assassin's Creed*: *Odyssey*, as detailed in the literature review. Similarly, should a game contain loot boxes, it would also be coded as containing 'pay to win' or 'cosmetic' microtransactions on the basis of whether those loot boxes contained cosmetic rewards or pay to win rewards. *NBA2K18*, for example, whose randomised 'player packs' contain new basketballers that give gamers an in-game advantage, would be coded as containing both pay to win and loot boxes.

A final note should be made regarding the time at which the presence or absence of these features was recorded. The presence or absence of all of the above features was coded on the basis of those features *currently appearing in-game at the time of analysis*. It was deemed infeasible to consistently determine whether games had added or removed these features at any point during the period under study (2012–2019). As covered in our discussion, some games may have inserted or removed loot boxes, pay to win microtransactions, or cosmetic microtransactions during the period. This is considered in our discussion.

The presence of each of the features outlined above were measured by having two researchers separately code each game for their presence or absence. A single illustrative example of *Counter-Strike* was provided as an exemplar at the beginning of the coding process. Following this, coders explored the presence or absence of relevant in-game features primarily through a combination of reading the game developer's documentation and descriptions, and searching for other information regarding the games online such as in forum posts. If this was insufficient, researchers engaged in watching videos of others playing the games and, as a last resort, playing the games in question themselves.

An initial round of coding resulted in near-perfect agreement between coders when it came to the presence of loot boxes (97%, Cohen's Kappa = 0.90). However, there was only substantial agreement when it came to the presence of pay to win (85.5%, Cohen's Kappa = 0.66) and cosmetic microtransactions (84.6%, Cohen's Kappa = 0.68).

Disagreement between human coders is an extremely common feature of any reliability analysis. Indeed, as noted in a standard textbook on the topic, for many researchers, a raw agreement rate of 80–90% is taken as sufficient in a variety of contexts [27].However, given the importance of a reliable coding scheme to our analyses, we resolved to be more stringent. Cohen's Kappa measures the degree of agreement between coders when chance is taken into account, and is the most commonly used way to measure inter-coder reliability. A Kappa statistic of greater than or equal to 0.81 is categorised according to multiple common benchmarking schemes as representing 'near perfect agreement' [28]. As in previous work on similar topics (e.g. [12]), it was determined that after achieving a minimum acceptable Kappa level, any remaining disagreements between coders would then be resolved through dialogical intersubjectivity to yield a dataset whose accuracy we were confident in.

Disagreements in coding were first discussed before re-coding the data. From these discussions, it emerged that disagreements in coding may have been due to a lack of clarity about whether downloadable content (DLC) such as expansion packs should be classified as either pay to win or cosmetic microtransactions. This is a subtle point. The simulation game *Farming Simulator 15*, for example, has 4 DLC releases that contain new branded machinery that may minorly improve a player's farming capabilities. Should this be classed as a game with pay to win microtransactions?

In order to resolve this, it was agreed that cosmetic and pay to win microtransactions would be classified as in-game items and rewards that are purchasable with real-world money *but do not add substantial additional game content*. This was undertaken in order to distinguish as best as possible between the addition of small amounts of additional content via microtransaction, and the offer to purchase substantial video game expansion packs such as in *Skyrim*. For example, the Echoes of Auriga Pack in *Endless Legends* may give the player in-game skins such as the Drum of Gios. However, it also comes with a substantial additional content in the form of a new soundtrack, and thus was not coded as a cosmetic microtransaction.

Every game in the dataset was then recoded separately by both coders using this new definition. This round of coding led to near-perfect agreement for both pay to win (96.5%, Cohen's Kappa = 0.91) and cosmetic microtransactions (96.3%, Cohen's Kappa = 0.92). Eleven games

remained uncoded at this point. These were either demos, test servers, or other non-game products (e.g. an SDK).

Both coders then met and discussed the remaining games on which their codes conflicted. The resolution of these cases via dialogic intersubjectivity led to perfect agreement, and a final dataset of games annotated with the presence of both loot boxes, pay to win features, and cosmetic microtransactions.

Overall, 463 games were included in the final dataset after removing the eleven that could not be categorised. There were 75 games with loot boxes, 388 games without them, and 11 games that could not be categorised. There were 135 games with pay to win microtransactions, 328 games without them, and 11 games that could not be categorised. There were 203 games with cosmetic microtransactions, 260 games without them, and 11 games that could not be categorised.

The presence of both multiplayer and co-operative features were additionally measured for each of these games. Analysis of these features is not presented here.

Changes in exposure to loot boxes, pay to win, and cosmetic microtransactions was measured by first recording the number of players of each game under test for each of the 3,319 measured days from 22$^{nd}$ March 2010 to 22$^{nd}$ April 2019. This was accomplished by extracting the complete history for each game from the *SteamDB* website. Any missing days were filled in via linear interpolation. The total number of players was summed for each day. The number of players of games with each specific feature on each of these days was then calculated. This figure was then divided by the total number of players overall for that day, and multiplied by 100 to yield a percentage measure of exposure.

## Statistical analysis

Changing trends in video game features were explored using joinpoint regression. Joinpoint regression is a technique for procedurally fitting a segmented regression model to trend data in order to identify points in a dataset at which a trend changes [29]. It begins by fitting a linear model to the dataset under test, and then iteratively tests whether the segmentation of this model via one or several 'joinpoints' leads to an improvement in overall fit. Joinpoint regression is suitable for the analysis of time series data, and commonly used to analyse change in trends over time. It is most commonly used in the analysis of changes in cancer rates over time. However, it has been used for analysing changes in trends as diverse as sales of pipe tobacco [30]; suicide rates [31]; fatal car crashes [32]; workforce growth [33]; and the prevalence of coronary heart disease [34].

Joinpoint regressions can be computationally expensive, and data were therefore transformed into weekly means in order to make analysis tractable. The National Cancer Institute's Joinpoint Regression Program Version 4.7.0.0 was used for these analyses. Due to the serial nature of the data, adjustments for autocorrelation were made according to [29]. Model selection was conducted by measuring the fit of each model via the calculation of BIC3, a variant of the Bayesian Information Criterion [35, p. 3]. In order to prevent the development of an overfitted model, we elected for a maximum of three joinpoints to be fit to the data, and for a minimum of 8 weeks to occur between joinpoints.

## Results

### Changes in exposure to loot boxes

Exploratory joinpoint regression was first carried out on the relationship between time and the percentage of individuals in the sample who played games which featured loot boxes. Exposure to loot boxes was initially estimated at 4.2% of the sample in 22$^{nd}$-26$^{th}$ March 2010, rising to 71.2% of the sample by 16th-22nd April 2019. Results indicated that the best-fitting model (BIC3 = 2.63) contained two joinpoints: 1$^{st}$-8$^{th}$ January 2012, and 12$^{th}$-19$^{th}$ March 2014.

Exposure first increased at an average annual rate of 5.3%, from 4.2% at the beginning of observation to 14.0% in the period of 1st-8th January 2012 (β = 0.10, t = 2.82, p = 0.004). At this point, the trend increased significantly in steepness (change in β = 0.28, t = 6.50, p<0.001) to an average annual increase of 20.3% (β = 0.39, t = 15.54, p<0.001). Finally, at the second inflection point during 12th-19th March 2014, exposure was estimated at 59.4%. At this point, the trend in the data became significantly more shallow (change in β = -0.34, t = -12.96, p<0.001). This led to a more gradual rise in exposure to 71.2% by 16th-22nd April 2019 at an average annual increase of 2.0% (β = 0.04, t = 4.78, p<0.001).

### Changes in exposure to pay to win microtransactions

Joinpoint regression was then carried out on the relationship between time and the percentage of individuals in the sample who played games with pay to win features. Exposure to pay to win features was initially estimated at 5.0% of the sample, rising to 15.9% of the sample by 16th-22nd April 2019. Results indicated that the best-fitting model (BIC3 = 1.51) contained a single joinpoint during 12th-19th November 2015.

Exposure first increased at an average annual rate of 2.1%, from 5.0% during 22nd-26th March 2010 to 17.3% during 12th-19th November 2015 (β = 0.04, t = 10.12, p<0.001). At the inflection point of 12th-19th November 2015, this trend decreased significantly in steepness (change in β = -0.04, t = -5.40, p<0.001) to an average annual rate that was not significantly different from zero (β = -0.008, t = -0.99, p = 0.32).

### Changes in exposure to cosmetic microtransactions

Joinpoint regression was finally carried out on the relationship between time and the percentage of individuals in the sample who played games which featured cosmetic microtransactions. Exposure to cosmetic microtransactions was initially estimated at 8.3% of the sample during 22nd-26th March 2010, rising to 85.8% of the sample by 16th-22nd April 2019. Results indicated that the best-fitting model (BIC3 = 2.34) contained two joinpoints: 12th-19th February 2012, and 20th-27th August 2013.

Exposure first increased at an average annual rate of 7.7%, from 8.3% at 22nd-26th March 2010 to 23.4% at 12th-19th February 2012 (β = 0.149, t = 5.13, p<0.001). At the first inflection point during 12th-19th February 2012, this trend increased significantly in steepness (change in β = 0.40, t = 9.02, p<0.001) to an average annual increase of 28.9% (β = 0.555, t = 16.18, p<0.001), leading to an estimated exposure of 67.8% during 20th-27th August 2013. At this point, the trend in the data became significantly more shallow (change in β = -0.49, t = -14.15, p<0.001). This led to a more gradual rise in exposure to 85.8% at 16th–22nd April 2019 at an average annual increase of 3.1% (β = 0.06, t = 8.84, p<0.001).

The resulting models from all joinpoint regression analyses are shown below as Fig 1.

## Discussion

These results corroborate reports of an overall growth in loot boxes and cosmetic microtransactions in the period 2010–2019. At the beginning of the period, only a small minority of gamers were exposed to these features: 5.3% and 8.3% of the sample respectively. However, by the end of the studied period, the majority of gamers were playing games that featured both loot boxes (71.2%) and cosmetic microtransactions (85.8%). This does not contradict statements by games industry representatives that loot boxes only appear in a minority of games: After all, a mere 75 of the 463 games analysed during this study contained loot boxes. However, they do suggest that the games which do contain loot boxes such as *DOTA 2* and *Player*

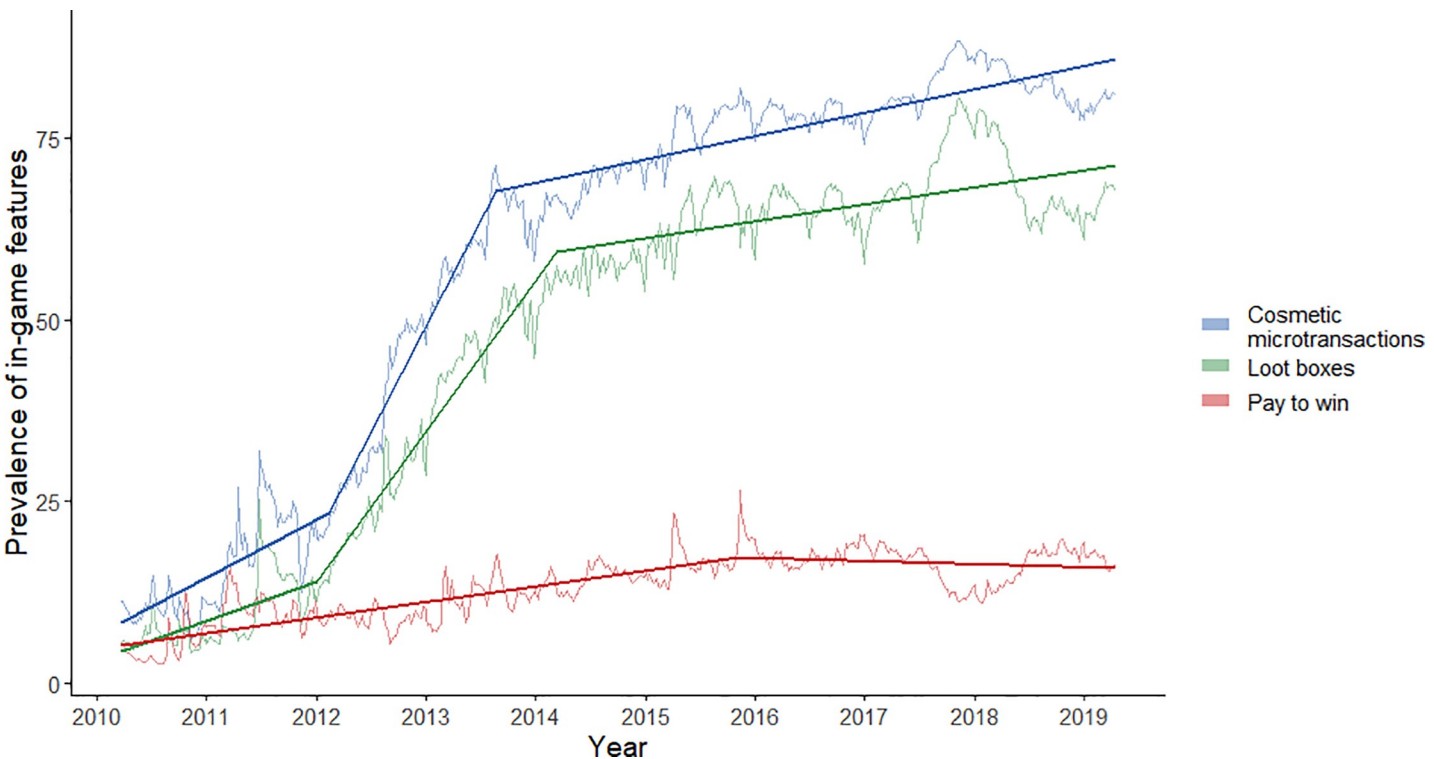

**Fig 1. Time series graph showing the percent of the sample playing games with each relevant feature during the period under test.** Models produced by three separate joinpoint regression analyses are superimposed on the graph as lines on top of each relevant time series.

*Unknown's Battlegrounds* may be so popular that, whilst the minority of games may have loot boxes, the majority of gamers are exposed to this feature.

It is important to note that the data under test also provides no evidence of a diminishment in the exposure to either loot boxes or cosmetic microtransactions: None of the regression analyses within any joinpoint model contained a negative coefficient. However, they do suggest that the growth of both cosmetic microtransactions and loot boxes have decelerated in recent years.

The majority of growth in exposure to loot boxes was modelled as taking place between January 2012 and March 2014. During this period exposure to loot boxes increased at a rate of 20.38% per year to a point where more than half of the sample played games with loot boxes. Similarly, between February 2012 and August 2013, exposure to cosmetic microtransactions increased at a rate of 28.9% per year to the point where more than two-thirds of the sample played games with cosmetic microtransactions. However, immediately after these rapid periods of growth, the increase in exposure to both these features dropped significantly to relatively low rates: 2.0% and 3.1% per year respectively. These low rates remained in place for the subsequent 5–6 years.

Some might be surprised by the growth in loot box exposure occurring as early as 2012–2014. They may assume that increases in loot box exposure occurred much later in time. For example, in [16], researchers state that "there were as many games released with loot boxes in 2016–2017 as there were before this time", suggesting that loot box exposure may similarly experience rapid increases after 2016. The novelty of the result observed here may be due to two factors. The first is that we are measuring exposure rather than prevalence: It may well be the case that changes in the number of *games* that contain loot boxes occur differently to changes in the number of *players* who are exposed to loot boxes.

However, another explanation for this difference in analysis is possible: data quality. The statement made above was based on scrutinising a list of games containing loot boxes that was collated and sourced from the video game journalism site *Giant Bomb*. Said data is subject to various forms of bias and inaccuracy. Furthermore, the underlying processes that are used to generate said data are not available for public scrutiny, and may not conform to scientific ideals. The data presented here is superior in this regard. Further work to determine the changing prevalence rates (as well as exposure rates) of loot boxes over time are necessary.

Exposure to pay to win microtransactions appeared to change in a somewhat different manner to the features outlined above. Whilst loot boxes and cosmetic microtransaction growth was characterised by a sharp increase leading to a slow period of gradual growth, pay to win microtransactions did not experience a similar temporary acceleration. Instead, exposure was modelled as rising at a gradual rate of only 2.1% per year from the beginning of observation until 12th-19th November 2015, at which point this rate declined ($p<0.001$) to an increase that was not significantly different from zero ($p = 0.32$). Consequently, by the end of the sampled period, only 15.9% of the sample were playing games that featured pay to win microtransactions.

## Limitations

The analyses presented here are limited in several ways. The dataset used captures the data of a large number of individuals: Indeed, an average of over 4 million players were recorded each day within our dataset by the conclusion of the studied time period. However, it is important to note that this data represents the players of only the 463 most popular games on *Steam*. The data of all less-popular games are is therefore not included in this dataset, and it is likely that these games may have a different distribution of features to the most popular games on the market.

Additionally, each game was coded as containing a specific feature if it contained that feature at the time of coding. Theoretically, a game may have only introduced a feature such as cosmetic microtransactions in 2017 or 2018. Yet, when coding took place, all datapoints for that game would be coded as coming from a game which contained such a feature. If this is the case, the models produced below could underestimate the size of increases in exposure within the sample. Furthermore, it is also possible that games in the sample had previously contained loot boxes, and then subsequently removed them. These games would be coded as not containing loot boxes, and their presence in the sample might lead to the overestimation of increases in exposure to loot boxes.

Finally, and most importantly, this dataset consists only of information about desktop games available via the *Steam* marketplace. It is therefore unable to provide information about the exposure to cosmetic microtransactions, loot boxes, and pay to win features on other platforms such as mobile devices.

One must also note that this data cannot make any claims about the number of players who actually purchased microtransactions of any kind; rather, it speaks to the frequency with which these features appeared in popular games, and the proportion of gamers who are exposed to these features in the games they play.

A final limitation of this study concerns the joinpoint analysis that was used. Joinpoint analyses are able to establish both when changes in the slope of a regression line occurred, and the steepness of a slope after each change-point. This makes them appropriate to the aims of this project: They allow researchers to understand how exposure to various in-game features has changed in specific ways over the past decade. Here, for example, they are used to estimate a rate of increase in exposure between specific dates. However, their utility in understanding

other features of exposure is limited: They are not commonly used as predictive models to understand future levels of exposure; and their ability to estimate the shape of trends is limited: They cannot be used to understand if, for example, increases in exposure fit an exponential curve. These are interesting and important analyses for future researchers to conduct. All data associated with our analyses are publicly available, and it is our hope that future work will address these questions.

Similarly, the process of joinpoint regression necessitates researchers defining specifc apriori constraints on their statistical models. For example, the regression undertaken here was constrained to contain a maximum of three joinpoints. It should be noted that these constraints were informed by standard practice in the literature: For example, it is a common strategy to allow a maximum of three joinpoints to occur within a model, presumably in order to resolve tensions between computational efficiency, overfitting, and model accuracy [36–38]. A sceptic may note that this analytic strategy may allow for situations of analytic flexibility: One might receive different results by specifying four, or five, or two joinpoints. Further work may focus on determining the impact of model constraint decisions on analytic outcomes: Our data is open and available with no reservations should other parties wish to reanalyse it for this purpose, via, for example, a form of multiverse analysis.

## Conclusions

The exploratory analysis presented above suggests that pay to win microtransactions continue to be an uncommon feature of desktop video games. Increases in exposure to this feature appeared to only gradually rise from 2010 onwards, and to plateau in 2015, leading to relatively low levels of exposure in 2019.

By contrast, cosmetic microtransactions and loot boxes appear to be present in games played by the majority of desktop gamers within the sample. Over 70% of gamers played a game with loot boxes in by the end of the studied period; over 80% played a game with cosmetic microtransactions. This increase in exposure does not appear recent: Indeed, the data suggests that these features may have risen to a dominant position in desktop games as early as 2014.

Academics and policymakers have expressed interest and concern in the potential consequences of the incorporation of the features outlined above in modern video games. Recent reports have suggested that loot boxes may recently have experienced either a decline in popularity, or a rise in popularity. This study instead suggests that, at least on desktop platforms, gamers experienced a sudden increase in exposure to both loot boxes and cosmetic microtransactions during approximately 2012–2014, followed by a period of steady and gradual growth.

## Author Contributions

**Conceptualization:** David Zendle.

**Data curation:** David Zendle, Rachel Meyer, Nick Ballou.

**Formal analysis:** David Zendle, Rachel Meyer, Nick Ballou.

**Methodology:** David Zendle, Rachel Meyer, Nick Ballou.

**Writing – original draft:** David Zendle.

**Writing – review & editing:** Rachel Meyer, Nick Ballou.

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
