## [Decision Letter · Decision Letter 0]

28 Jan 2020

PONE-D-19-30715

The changing face of desktop video game monetisation: An exploration of trends in loot boxes, pay to win, and cosmetic microtransactions in the most-played Steam games of 2010-2019

PLOS ONE

Dear Dr. Zendle,

Thank you for submitting your manuscript to PLOS ONE. After careful consideration, we feel that it has merit but does not fully meet PLOS ONE’s publication criteria as it currently stands. Therefore, we invite you to submit a revised version of the manuscript that addresses the points raised during the review process.

Editor comments:

I have now the reviews on your manuscript from three experts on the topic. In spite of the differences in the tone of their general assessments, the three reports are mostly coincident in content, and converge in presenting some concerns that should be addressed before the paper is considered suitable for publication.

The first of them has to do with the very definition of pay-to-win, cosmetic microtransactions, and loot boxes. As pointed out by R2 and R3, loot boxes can contain items that can confer or not competitive advantage to the player who purchases them, so some overlap exists between loot boxes and the other two categories. Please provide the necessary information on how this overlap was resolved (as well as on the other definitional aspects mentioned by the three reviewers).

The second important issue regards the use of “prevalence”. In my view (and R3’s), prevalence seems to refer to the number of games that contain a feature, whereas “exposure” better describes how many people are exposed to such a feature. Actually, I completely agree with R3 that the two analyses are interesting enough to be presented together. (In terms of policy, the implications of an increasing number of games including certain feature are very different to the ones of an increasing number of people purchasing and playing games including that feature). And, in any case, a systematic and transparent terminology should be used throughout the manuscript.

In relation to this (sorry if I missed something here), I guess there are games for which features of interests (i.e. microtransactions) were added or removed during the period under scrutiny. Did that actually occur? And, if it did, how was it recorded and taken into account for analyses?

Finally, although not mentioned by the reviewers, I suggest to better justify the statistical approach. Why was joinpoint analysis used instead of trying to fit some continuous function? As far as I know, joinpoint regression is aimed at detecting time points in which trends change, so they can be interpreted in relation to certain events (e.g. policy changes, or new legislations). I understand that trying explain post-hoc what happened at such point would be highly speculative. Yet, in the absence of prior hypotheses regarding trend changes, the use of this analysis (and the a priori decision to limit the number of joinpoints to 3) may seem arbitrary.  

We would appreciate receiving your revised manuscript by Mar 13 2020 11:59PM. To enhance the reproducibility of your results, we recommend that if applicable you deposit your laboratory protocols in protocols.io, where a protocol can be assigned its own identifier (DOI) such that it can be cited independently in the future. For instructions see: http://journals.plos.org/plosone/s/submission-guidelines#loc-laboratory-protocols

We look forward to receiving your revised manuscript.

Kind regards,

José C. Perales

Academic Editor

PLOS ONE

Journal Requirements:

2. In your ethics statement in the online submission information and in the manuscript, please include the full name of the department that issued the ethics waiver.

Reviewers' comments:

Reviewer's Responses to Questions

**Comments to the Author**

1. Is the manuscript technically sound, and do the data support the conclusions?

Reviewer #1: Yes

Reviewer #2: Partly

Reviewer #3: Partly

2. Has the statistical analysis been performed appropriately and rigorously? 

Reviewer #1: Yes

Reviewer #2: I Don't Know

Reviewer #3: Yes

3. Have the authors made all data underlying the findings in their manuscript fully available?

Reviewer #1: Yes

Reviewer #2: No

Reviewer #3: Yes

4. Is the manuscript presented in an intelligible fashion and written in standard English?

Reviewer #1: Yes

Reviewer #2: Yes

Reviewer #3: Yes

5. Review Comments to the Author

Reviewer #1: This was a very clearly written article and the data with an interesting application of joinpoint regression that I had not seen before. I have only a couple of observations:

1) In the 4 paragraph (introduction) you note that "Loot Boxes are items or bonuses in video games that players can buy with real-world money...." However, Loot Boxes can also refer to in-game items that are rewarded from gameplay or non-monetary points accumulated in a game. It would be helpful if you simply noted your more restrictive definition of loot boxes.

2) When quoting growth rates throughout the document, you use 2 significant digits (e.g., 5.31%). Given the nature of the analyses, I don't think the second digit is at all reliable, and thus indicates a level of precision in your estimates that likely does not exist. I suggest - and this is only a suggestion - that it would be more appropriate to only use 1 significant digit (e.g., 5.3%).

Reviewer #2: The paper seeks to examine the prevalence rate of two types of purchasable virtual items, so-called cosmetic and pay-to-win items, and a monetization technique colloquially known as loot boxes.

The paper examines an interesting and important topic in game studies. However, I have some major concerns with the paper, one is methodological and one in ontological. First of all, the authors several times distinguish between cosmetic microtransactions, pay-to-win microtransactions, and loot boxes. This gives the impression that the authors understand these to be distinct objects. However, since loot boxes can contain both cosmetic and pay-to-win objects this is misleading to the reader. Pay-to-win items and cosmetic items, it seems to me that loot boxes are not as much items as a procedure for doling out rewards or products in a transaction. I'd like the authors to acknowledge and reflect on this. Or alternatively tell me why I am wrong in my view.

Second of all, it is not made clear how the researchers have obtained a deep understanding of the vast number of games that they have scored. I am not convinced by the paper that the authors are familiar enough with the games that they score for the scores to be meaning full. Even if every game can be played through relatively quickly it would still be a massively time-consuming endeavor to play through all of them. Is the rating process based on firsthand knowledge of the games or is there a database somewhere that they are using?

Thirdly, I think the authors should be more precise in their definition of what loot boxes are. If we take packs of footballers purchased by the player in Fifa as an example, I’d like to know if the authors categorize these as loot boxes or not. The packs are the same in all countries, but in some they can be purchase with real money and in some they cannot. Following the authors definition these same packs could then be categorized as both loot boxes and not loot boxes. This, I assume is true of all games. This, I think, requires the authors to be much more transparent about their methodology and results.

Two minor problems here at the end:

1) In the introduction it seems that the authors forgot the part of history where games as a service emerged (i.e. subscription-based monetization strategies).

2) The misunderstanding about DLC and whether to code that as pay-to-win or cosmetic microtransactions leave me questioning the competence of the raters. The paper reads as follows: “After this round of coding, it emerged that disagreements in coding may have been due to a lack of clarity about whether downloadable content (DLC) such as expansion packs should be classified as either pay to win or cosmetic microtransactions. In order to resolve this, it was agreed that cosmetic and pay to win microtransactions would be classified as in-game items and rewards that are purchasable with real-world money but do not add substantial additional game content.”

That such an argument could even take place makes me question the raters game literacy or knowledge of games, which is problematic because the raters rating of a massive amount of games is the foundation of the paper.

All in all, I think it is an interesting paper, and I wish I could have given it a better review, but I think the paper is not fit for publication in its current state. I tried to access the data that is made available online, but found the data in the files to be really hard to parse. Maybe the files are just corrupt or otherwise damaged.

Reviewer #3: This is a well-written manuscript reporting an interesting piece of research. I enjoyed reading it, and believe these data will make a valuable contribution to the literature. The introduction provides a nice, concise introduction to the issue based on the extant literature, and a solid rationale for this work. Essentially, much is being of made of the potential risks associated with predatory monetization practices, but the actual prevalence of these practices is disputed. A systematic empirical examination of the prevalence of monetization mechanisms in video games will help researchers, policy-makers, and gamers to better appreciate the potential for risk posed by these mechanisms. As I say, I think this research is valuable. However, below I outline some areas where, based on my reading, the manuscript might benefit from further clarification. Importantly, I don’t see any of these issues as fatal.

Introduction

The taxonomies presented to characterize micro-transactions could be clearer. There seem to be multiple dimensions at play here. First, are the virtual items obtained purely aesthetic, or do they alter gameplay and confer an in-game advantage? Second, are they purchased directly, or obtained randomized reward mechanisms (e.g., loot boxes)? Although items that provide competitive boosts and are purchased directly seem to clearly fall under the “pay-to-win” heading, I’m not sure items obtained from loot boxes would, even if they confer a competitive advantage. Admittedly, this might be a purely subjective distinction. However, I wonder if both the function of the item and the method of acquisition are likely to be important to understanding whether the microtransaction in question constitutes a pay-to-win mechanism. Similarly, when considering “gambling-like entrapment”, it seems important to separate the nature of the item from the nature of the reward delivery mechanism (i.e., randomized outcome vs. direct purchase). This becomes clearer in the “present research” section – where microtransactions are characterized as loot boxes, pay to win, or cosmetic – but I think this issue could be a little more clearly distilled in the introduction.

Method

* The method for acquiring data was generally clearly articulated, and the criteria for selecting the games included seems sensible. How did reviewers code for the presence or absence of microtransactions? Did they source data from Steam, or from game review platforms, etc.?

* Interested readers might benefit from further technical details (e.g., in supplemental files) relating to how data were obtained from the SteamDB website.

Results

* I’m not familiar with joinpoint regression, but the analysis was described clearly from a conceptual standpoint and seems appropriate for the task at hand.

* Having said that, I was a little surprised by the analysis reported. To estimate the prevalence of microtransactions in games and changes in this prevalence over time, I expected to see a comparison of the percentage of popular games featuring and not featuring microtransactions on a year-by-year basis. Admittedly, this is a comparatively unsophisticated analysis, but it seems to get at the prevalence issue. The analyses reported seem to speak to a separate issue: The percentage of players who are playing games with microtransactions. This is potentially an interesting question in itself, and may relate to prevalence of engagement with these mechanisms (though it is a proxy at best: as the authors note in the discussion, playing a game with microtransactions and engaging with microtransactions are not the same thing), but it doesn’t seem to speak to the prevalence of microtransactions themselves. I might be wrong here but, if so, I think the authors could be clearer about how the chosen analysis speaks to the prevalence of these mechanisms in the manner discussed in the introduction.

* Nonetheless, the analyses presented are clearly reported, and augmented nicely by the data in Figure 1.

Discussion

* Again, I’m not sure these analyses show “an overall growth in loot boxes and cosmetic microtransactions”. Instead, they seem to show an increase in the percentage of players in the sample who played games featuring microtransactions. Again, this seems to speak to player engagement with games housing these features, rather than the prevalence of the features themselves. This is hinted at by the authors acknowledgment that, even though most of the gamers in the sample were playing games that included loot boxes, only 75 of 463 games reviewed contained loot boxes. Again, apologies if I’ve misunderstood what these data are telling me.

* Nonetheless, the more basic interpretations of the data are sound: engagement with games that feature microtransactions and loot boxes increased notably over the period of time for which data were analyzed.

* It interesting to note that, perhaps counter to many people’s perceptions, the most rapid period of growth (according to the data presented here) was between 2012 and 2014 (cf. the more recent examples that have captured public and media attention). Though, once again, this potentially highlights a divergence from the issue of prevalence: Drummond & Sauer (2018) note that were as many games released with loot boxes in 2016–2017 as there were before this time suggesting that, for console games at least, the prevalence of loot box mechanisms increased most rapidly in that time period (cf. the 2012-2014 evidenced in the current data). I wonder if this difference reflects (a) differences in console vs. desktop gaming or (b) differences in the way prevalence is estimated?

6. PLOS authors have the option to publish the peer review history of their article (what does this mean?). If published, this will include your full peer review and any attached files.

Reviewer #1: No

Reviewer #2: No

Reviewer #3: No

---

## [Author Response · Author response to Decision Letter 0]

2 Mar 2020

Dear Dr. Perales,

We thank both yourself and the reviewers for their detailed remarks. We provide a point-by-point response to each comment below.

“I have now the reviews on your manuscript from three experts on the topic. In spite of the differences in the tone of their general assessments, the three reports are mostly coincident in content, and converge in presenting some concerns that should be addressed before the paper is considered suitable for publication.

The first of them has to do with the very definition of pay-to-win, cosmetic microtransactions, and loot boxes. As pointed out by R2 and R3, loot boxes can contain items that can confer or not competitive advantage to the player who purchases them, so some overlap exists between loot boxes and the other two categories. Please provide the necessary information on how this overlap was resolved (as well as on the other definitional aspects mentioned by the three reviewers).”

We have extended our method section to describe how overlap between loot boxes, pay to win, and cosmetic microtransactions was considered within our analysis. Other definitional questions touched on by the editor have been addressed through additions to our method and introduction. These are detailed below in the rest of this rebuttal, alongside relevant reviewer comments.

“The second important issue regards the use of “prevalence”. In my view (and R3’s), prevalence seems to refer to the number of games that contain a feature, whereas “exposure” better describes how many people are exposed to such a feature. Actually, I completely agree with R3 that the two analyses are interesting enough to be presented together. (In terms of policy, the implications of an increasing number of games including certain feature are very different to the ones of an increasing number of people purchasing and playing games including that feature). And, in any case, a systematic and transparent terminology should be used throughout the manuscript.”

We have adjusted our terminology throughout the manuscript in line with this comment. We now consistently refer to exposure as the number of gamers playing games with a feature; and prevalence as the number of games with that feature.

“In relation to this (sorry if I missed something here), I guess there are games for which features of interests (i.e. microtransactions) were added or removed during the period under scrutiny. Did that actually occur? And, if it did, how was it recorded and taken into account for analyses?”

The addition or removal of features during the time-period under test was not taken into account in our analyses. Our method has been augmented to clarify why this was the case; our discussion has described how this may affect our analyses, and suggested further work on the topic.

“Finally, although not mentioned by the reviewers, I suggest to better justify the statistical approach. Why was joinpoint analysis used instead of trying to fit some continuous function? As far as I know, joinpoint regression is aimed at detecting time points in which trends change, so they can be interpreted in relation to certain events (e.g. policy changes, or new legislations). I understand that trying explain post-hoc what happened at such point would be highly speculative. Yet, in the absence of prior hypotheses regarding trend changes, the use of this analysis (and the a priori decision to limit the number of joinpoints to 3) may seem arbitrary.” 

The ‘limitations’ subsection of our manuscript has been extended to better motivate the use of joinpoint analysis and to highlight the open nature of the dataset for future re-analysis and investigation.

 “Reviewer #1: This was a very clearly written article and the data with an interesting application of joinpoint regression that I had not seen before. I have only a couple of observations:

1) In the 4 paragraph (introduction) you note that "Loot Boxes are items or bonuses in video games that players can buy with real-world money...." However, Loot Boxes can also refer to in-game items that are rewarded from gameplay or non-monetary points accumulated in a game. It would be helpful if you simply noted your more restrictive definition of loot boxes.”

Our manuscript has been altered to highlight the definition of loot boxes used here: It should be noted that this is the definition used by the UK government at a recent Select Committee inquiry, so it may come into more common usage.

“2) When quoting growth rates throughout the document, you use 2 significant digits (e.g., 5.31%). Given the nature of the analyses, I don't think the second digit is at all reliable, and thus indicates a level of precision in your estimates that likely does not exist. I suggest - and this is only a suggestion - that it would be more appropriate to only use 1 significant digit (e.g., 5.3%).”

We have adjusted our manuscript as suggested by the reviewer, and 1 significant digit is now used throughout

“Reviewer #2: The paper seeks to examine the prevalence rate of two types of purchasable virtual items, so-called cosmetic and pay-to-win items, and a monetization technique colloquially known as loot boxes.

The paper examines an interesting and important topic in game studies. However, I have some major concerns with the paper, one is methodological and one in ontological. First of all, the authors several times distinguish between cosmetic microtransactions, pay-to-win microtransactions, and loot boxes. This gives the impression that the authors understand these to be distinct objects. However, since loot boxes can contain both cosmetic and pay-to-win objects this is misleading to the reader. Pay-to-win items and cosmetic items, it seems to me that loot boxes are not as much items as a procedure for doling out rewards or products in a transaction. I'd like the authors to acknowledge and reflect on this. Or alternatively tell me why I am wrong in my view.”

The manuscript has been augmented to incorporate a more clear delineation between loot boxes, pay to win microtransactions, and cosmetic microtransactions. The definitional points made by R2 have been taken into account.

“Second of all, it is not made clear how the researchers have obtained a deep understanding of the vast number of games that they have scored. I am not convinced by the paper that the authors are familiar enough with the games that they score for the scores to be meaning full. Even if every game can be played through relatively quickly it would still be a massively time-consuming endeavor to play through all of them. Is the rating process based on firsthand knowledge of the games or is there a database somewhere that they are using?”

The manuscript has been updated to more fully describe the process that the authors went through in order to establish the presence of loot boxes in each game.

“Thirdly, I think the authors should be more precise in their definition of what loot boxes are. If we take packs of footballers purchased by the player in Fifa as an example, I’d like to know if the authors categorize these as loot boxes or not. The packs are the same in all countries, but in some they can be purchase with real money and in some they cannot. Following the authors definition these same packs could then be categorized as both loot boxes and not loot boxes. This, I assume is true of all games. This, I think, requires the authors to be much more transparent about their methodology and results.”

The definition of loot boxes used in this study has been further highlighted, and a brief discussion of the limitations and advantages of the approach to definition used has been incorporated into the manuscript.

“Two minor problems here at the end:

1) In the introduction it seems that the authors forgot the part of history where games as a service emerged (i.e. subscription-based monetization strategies).”

We have augmented our introduction to briefly mention subscription-based monetisation

“2) The misunderstanding about DLC and whether to code that as pay-to-win or cosmetic microtransactions leave me questioning the competence of the raters. The paper reads as follows: “After this round of coding, it emerged that disagreements in coding may have been due to a lack of clarity about whether downloadable content (DLC) such as expansion packs should be classified as either pay to win or cosmetic microtransactions. In order to resolve this, it was agreed that cosmetic and pay to win microtransactions would be classified as in-game items and rewards that are purchasable with real-world money but do not add substantial additional game content.”

That such an argument could even take place makes me question the raters game literacy or knowledge of games, which is problematic because the raters rating of a massive amount of games is the foundation of the paper.

Disagreements between coders are extremely common during reliability analyses. We believe there is a danger in reflexively stigmatising such disagreements. We strongly believe that it is important to be transparent about these disagreements, in order to create a manuscript that most accurately reflects the underlying data and analysis. We have augmented our manuscript to describe how common this is, and hope that this discussion will convince the reviewer of the credibility of the research team.

“All in all, I think it is an interesting paper, and I wish I could have given it a better review, but I think the paper is not fit for publication in its current state. I tried to access the data that is made available online, but found the data in the files to be really hard to parse. Maybe the files are just corrupt or otherwise damaged.”

The data associated with this manuscript are freely available on the OSF repository associated with this manuscript, as are the exact scripts used to run the major analyses. We appreciate that these may be difficult to parse, and hence have included a brief document as a key in response to R2’s critique.

Reviewer #3: This is a well-written manuscript reporting an interesting piece of research. I enjoyed reading it, and believe these data will make a valuable contribution to the literature. The introduction provides a nice, concise introduction to the issue based on the extant literature, and a solid rationale for this work. Essentially, much is being of made of the potential risks associated with predatory monetization practices, but the actual prevalence of these practices is disputed. A systematic empirical examination of the prevalence of monetization mechanisms in video games will help researchers, policy-makers, and gamers to better appreciate the potential for risk posed by these mechanisms. As I say, I think this research is valuable. However, below I outline some areas where, based on my reading, the manuscript might benefit from further clarification. Importantly, I don’t see any of these issues as fatal.

“The taxonomies presented to characterize micro-transactions could be clearer. There seem to be multiple dimensions at play here. First, are the virtual items obtained purely aesthetic, or do they alter gameplay and confer an in-game advantage? Second, are they purchased directly, or obtained randomized reward mechanisms (e.g., loot boxes)? Although items that provide competitive boosts and are purchased directly seem to clearly fall under the “pay-to-win” heading, I’m not sure items obtained from loot boxes would, even if they confer a competitive advantage. Admittedly, this might be a purely subjective distinction. However, I wonder if both the function of the item and the method of acquisition are likely to be important to understanding whether the microtransaction in question constitutes a pay-to-win mechanism.”

A common theme of comments from all reviewers is the need for a clearer discussion of what is specifically meant in this manuscript when a game is categorised as containing either pay to win, loot boxes, or cosmetic microtransactions. This point is well-taken, and a detailed discussion that addresses R3’s questions has been incorporated into the manuscript.

 Similarly, when considering “gambling-like entrapment”, it seems important to separate the nature of the item from the nature of the reward delivery mechanism (i.e., randomized outcome vs. direct purchase). This becomes clearer in the “present research” section – where microtransactions are characterized as loot boxes, pay to win, or cosmetic – but I think this issue could be a little more clearly distilled in the introduction.

The manuscript has been augmented with additional information clarifying the potential role of gambling-like entrapment in in-game spending.

“Method 

* The method for acquiring data was generally clearly articulated, and the criteria for selecting the games included seems sensible. How did reviewers code for the presence or absence of microtransactions? Did they source data from Steam, or from game review platforms, etc.?”

A more clear and detailed description of how each game was coded has been incorporated into our method section in response to reviewer feedback.

* Interested readers might benefit from further technical details (e.g., in supplemental files) relating to how data were obtained from the SteamDB website.

Further technical details have been incorporated into the manuscript about how data was collected from the SteamDB website. However, we feel that the reviewer may find our methods for data extraction somewhat plodding and less exciting than they may have imagined!

Results

* I’m not familiar with joinpoint regression, but the analysis was described clearly from a conceptual standpoint and seems appropriate for the task at hand.

“* Having said that, I was a little surprised by the analysis reported. To estimate the prevalence of microtransactions in games and changes in this prevalence over time, I expected to see a comparison of the percentage of popular games featuring and not featuring microtransactions on a year-by-year basis. Admittedly, this is a comparatively unsophisticated analysis, but it seems to get at the prevalence issue. The analyses reported seem to speak to a separate issue: The percentage of players who are playing games with microtransactions. This is potentially an interesting question in itself, and may relate to prevalence of engagement with these mechanisms (though it is a proxy at best: as the authors note in the discussion, playing a game with microtransactions and engaging with microtransactions are not the same thing), but it doesn’t seem to speak to the prevalence of microtransactions themselves. I might be wrong here but, if so, I think the authors could be clearer about how the chosen analysis speaks to the prevalence of these mechanisms in the manner discussed in the introduction.”

* Nonetheless, the analyses presented are clearly reported, and augmented nicely by the data in Figure 1.

Discussion

* Again, I’m not sure these analyses show “an overall growth in loot boxes and cosmetic microtransactions”. Instead, they seem to show an increase in the percentage of players in the sample who played games featuring microtransactions. Again, this seems to speak to player engagement with games housing these features, rather than the prevalence of the features themselves. This is hinted at by the authors acknowledgment that, even though most of the gamers in the sample were playing games that included loot boxes, only 75 of 463 games reviewed contained loot boxes. Again, apologies if I’ve misunderstood what these data are telling me.

The reviewer is correct in their inferences, and their points are well-taken. In response to this comment (and aligned editorial feedback), we have revised the manuscript to frame our analyses as a description of ‘exposure’ rather than ‘prevalence’.

“* Nonetheless, the more basic interpretations of the data are sound: engagement with games that feature microtransactions and loot boxes increased notably over the period of time for which data were analyzed.

* It interesting to note that, perhaps counter to many people’s perceptions, the most rapid period of growth (according to the data presented here) was between 2012 and 2014 (cf. the more recent examples that have captured public and media attention). Though, once again, this potentially highlights a divergence from the issue of prevalence: Drummond & Sauer (2018) note that were as many games released with loot boxes in 2016–2017 as there were before this time suggesting that, for console games at least, the prevalence of loot box mechanisms increased most rapidly in that time period (cf. the 2012-2014 evidenced in the current data). I wonder if this difference reflects (a) differences in console vs. desktop gaming or (b) differences in the way prevalence is estimated?”

The reviewer raises interesting points. A treatment of the points raised – including specific reference to Drummond and Sauer’s points - has been integrated into our discussion.

---

## [Decision Letter · Decision Letter 1]

11 Apr 2020

PONE-D-19-30715R1

The changing face of desktop video game monetisation: An exploration of exposure to loot boxes, pay to win, and cosmetic microtransactions in the most-played Steam games of 2010-2019

PLOS ONE

Dear Dr. Zendle,

Thank you for submitting your revised manuscript to PLOS ONE.

As you can see in the attached reports, the reviewers are mostly satisfied with the level of detail with which you have considered their comments and suggestions.

Only one of them still finds your definition of loot boxes not precise enough fot an academic context. Please try to address that remaining concern etither making minor changes in your manuscript or in your response letter. In principle, if I find your reply convincing enough, no further review rounds will be required.

We would appreciate receiving your revised manuscript by May 26 2020 11:59PM. To enhance the reproducibility of your results, we recommend that if applicable you deposit your laboratory protocols in protocols.io, where a protocol can be assigned its own identifier (DOI) such that it can be cited independently in the future. For instructions see: http://journals.plos.org/plosone/s/submission-guidelines#loc-laboratory-protocols

We look forward to receiving your revised manuscript.

Kind regards,

José C. Perales

Academic Editor

PLOS ONE

Reviewers' comments:

Reviewer's Responses to Questions

**Comments to the Author**

1. If the authors have adequately addressed your comments raised in a previous round of review and you feel that this manuscript is now acceptable for publication, you may indicate that here to bypass the “Comments to the Author” section, enter your conflict of interest statement in the “Confidential to Editor” section, and submit your "Accept" recommendation.

Reviewer #1: All comments have been addressed

Reviewer #2: (No Response)

Reviewer #3: (No Response)

2. Is the manuscript technically sound, and do the data support the conclusions?

Reviewer #1: Yes

Reviewer #2: Yes

Reviewer #3: Yes

3. Has the statistical analysis been performed appropriately and rigorously? 

Reviewer #1: Yes

Reviewer #2: I Don't Know

Reviewer #3: Yes

4. Have the authors made all data underlying the findings in their manuscript fully available?

Reviewer #1: Yes

Reviewer #2: Yes

Reviewer #3: Yes

5. Is the manuscript presented in an intelligible fashion and written in standard English?

Reviewer #1: Yes

Reviewer #2: Yes

Reviewer #3: Yes

6. Review Comments to the Author

Reviewer #1: My comments and suggestions have been adequately addressed. The problem with definitions of loot boxes has been addressed, and the suggestion to reduce significant digits to 1 was also accepted.

Reviewer #2: I thank the authors for their replies to our comments. I am satisfied that they have made the needed corrections in most cases. Furthermore, I am impressed that the researchers have downloaded and played such an enormous amount of games. Cudos.

The one exception is their definition of loot boxes, which is still think is too vague and ambiguous for an academic paper. The authors still do not account for the fact that the same loot box can be purchased with fiat currency or in-game currency accrued through play. Instead, they seem to argue that the same item (e.g. a card pack) in digital games are loot boxes if purchased with fiat currency and not loot boxes if purchased through play.

I am not convinced that the definition provided by the UK Parliament is of much use in an academic context. This would not be problematic if the authors did not liken loot boxes to slot machines. If the authors believe that digital collectible card games are gambling do the authors then also believe that non-digital collectible card games are gambling? I’m assuming this would go against the view of the UK Parliament.

I’m sure this line of argument seems pedantic to the authors, but I think it is disingenuous to imply that loot boxes are simply slot machines in games. Surely there are more nuances than that? Otherwise, how could so many other countries in the world come to the conclusion that loot boxes are in fact not gambling? Acknowledging the complexities of the issue would go a long way.

Reviewer #3: PONE-D-19-30715_R1

I was Reviewer 3 on the original manuscript.

The authors have been thorough in their response to the reviews. In particular, the changes to the introduction make for a clearer explanation of cosmetic and pay-to-win microtransactions, and their relationship to loot boxes. In general, as a reader, I appreciated the additional detail relating to how the data were obtained and how they were coded (even if this detail was not necessarily exciting). I liked the manuscript before, and I think it is even better now. It will make a valuable contribution to the literature.

I have nothing further to request from the authors.

As I side note, it was really useful to have the track-changes version of the document available to appreciate the scope of the revisions made.

7. PLOS authors have the option to publish the peer review history of their article (what does this mean?). If published, this will include your full peer review and any attached files.

Reviewer #1: No

Reviewer #2: No

Reviewer #3: No

---

## [Author Response · Author response to Decision Letter 1]

11 Apr 2020

Dear Dr. Perales,

We thank both yourself and the reviewers for their remarks. We provide a response to reviewer 2’s comment below.

Reviewer 2's comments

I thank the authors for their replies to our comments. I am satisfied that they have made the needed corrections in most cases. Furthermore, I am impressed that the researchers have downloaded and played such an enormous amount of games. Cudos.

The one exception is their definition of loot boxes, which is still think is too vague and ambiguous for an academic paper. The authors still do not account for the fact that the same loot box can be purchased with fiat currency or in-game currency accrued through play. Instead, they seem to argue that the same item (e.g. a card pack) in digital games are loot boxes if purchased with fiat currency and not loot boxes if purchased through play.

I am not convinced that the definition provided by the UK Parliament is of much use in an academic context. This would not be problematic if the authors did not liken loot boxes to slot machines. If the authors believe that digital collectible card games are gambling do the authors then also believe that non-digital collectible card games are gambling? I’m assuming this would go against the view of the UK Parliament.

I’m sure this line of argument seems pedantic to the authors, but I think it is disingenuous to imply that loot boxes are simply slot machines in games. Surely there are more nuances than that? Otherwise, how could so many other countries in the world come to the conclusion that loot boxes are in fact not gambling? Acknowledging the complexities of the issue would go a long way.

The reviewer makes several points here which we would like to respond to, as they contain some misinterpretation of the work presented in this paper. 

Firstly, reviewer 2 states that the definition of loot boxes that we use in our manuscript is ‘provided by’ the UK Parliament, and therefore is of little use in an academic context. However, they are incorrect. If they inspect our manuscript, they will see that this definition is taken from a recent article published in Addiction (a scholarly journal). As noted in the manuscript, this academic definition was also *used* by a UK Parliamentary Select Committee to define loot boxes for the purposes of making recommendations regarding regulation. We believe this uptake in both academic and governmental contexts suggests it may have some utility.

Secondly, the reviewer provides the following objection regarding our definition of loot boxes: “I think it is disingenuous to imply that loot boxes are simply slot machines in games”. We have not stated that loot boxes are ‘simply slot machines in games’ in our manuscript, and do not believe it to be the case. This paper does not engage in definitional work of that nature in any way. We are unclear where the reviewer believes we make this argument. The closest that we come to is the text quoted below:

As noted in [16], loot boxes share distinct similarities with gambling. Both when paying for a loot box and when putting money into a slot machine, individuals are wagering something of value on the chance hope of receiving something of greater value. This has led to concerns that engaging with loot boxes may lead to increases in gambling amongst gamers [17]. Evidence for this causal mechanism is unclear. Spending on loot boxes has been repeatedly linked to problem gambling. However, it is uncertain whether this is because loot boxes cause problem gambling, or whether it is because individuals with pre-existing gambling problems spend more money on loot boxes [18]–[20].

We find the text above uncontroversial and accurate. However, in order to assure that misinterpretations such as the one reviewer 2 makes do not occur, we have removed the following sentence from our manuscript: “Both when paying for a loot box and when putting money into a slot machine, individuals are wagering something of value on the chance hope of receiving something of greater value.” 

Thirdly, the reviewer states that they are unconvinced that our definition of loot boxes is of use in an academic context for the following reason: “This would not be problematic if the authors did not liken loot boxes to slot machines. If the authors believe that digital collectible card games are gambling do the authors then also believe that non-digital collectible card games are gambling?”. 

We are not entirely clear with what reviewer 2 is getting at here. They state that the authors ‘believe that digital collectible card games are gambling’. Yet nowhere in this manuscript have we classified loot boxes as ‘gambling’ or ‘not gambling’. The only paragraph that could even potentially be read this way is the one we quote above. As we have now removed this sentence, we hope that this potential misinterpretation may also now be resolved.

---

## [Editor Report · Decision Letter 2]

22 Apr 2020

The changing face of desktop video game monetisation: An exploration of exposure to loot boxes, pay to win, and cosmetic microtransactions in the most-played Steam games of 2010-2019

PONE-D-19-30715R2

Dear Dr. Zendle,

We are pleased to inform you that your manuscript has been judged scientifically suitable for publication and will be formally accepted for publication once it complies with all outstanding technical requirements.

With kind regards,

José C. Perales

Academic Editor

PLOS ONE
---

## [Editor Report · Acceptance letter]

28 Apr 2020

PONE-D-19-30715R2 

The changing face of desktop video game monetisation: An exploration of exposure to loot boxes, pay to win, and cosmetic microtransactions in the most-played Steam games of 2010-2019 

Dear Dr. Zendle:

I am pleased to inform you that your manuscript has been deemed suitable for publication in PLOS ONE. Congratulations! Your manuscript is now with our production department. 

With kind regards,

on behalf of

Dr. José C. Perales 

Academic Editor

PLOS ONE